# Bone Formation on Murine Cranial Bone by Injectable Cross-Linked Hyaluronic Acid Containing Nano-Hydroxyapatite and Bone Morphogenetic Protein

**DOI:** 10.3390/polym14245368

**Published:** 2022-12-08

**Authors:** Yuki Hachinohe, Masayuki Taira, Miki Hoshi, Wataru Hatakeyama, Tomofumi Sawada, Hisatomo Kondo

**Affiliations:** 1Department of Prosthodontics and Oral Implantology, School of Dentistry, Iwate Medical University, 19-1 Uchimaru, Morioka 020-8505, Japan; 2Department of Biomedical Engineering, Iwate Medical University, 1-1-1 Idaidori, Yahaba-cho 028-3694, Japan

**Keywords:** cross-linked hyaluronic acid, nano hydroxyapatite, bone morphogenetic protein, injection-type bone forming material, ectopic bone formation, bone augmentation

## Abstract

New injection-type bone-forming materials are desired in dental implantology. In this study, we added nano-hydroxyapatite (nHAp) and bone morphogenetic protein (BMP) to cross-linkable thiol-modified hyaluronic acid (tHyA) and evaluated its usefulness as an osteoinductive injectable material using an animal model. The sol (ux-tHyA) was changed to a gel (x-tHyA) by mixing with a cross-linker. We prepared two sol–gel (SG) material series, that is, x-tHyA + BMP with and without nHAp (SG I) and x-tHyA + nHAp with and without BMP (SG II). SG I materials in the sol stage were injected into the cranial subcutaneous connective tissues of mice, followed by in vivo gelation, while SG II materials gelled in Teflon rings were surgically placed directly on the cranial bones of rats. The animals were sacrificed 8 weeks after implantation, followed by X-ray analysis and histological examination. The results revealed that bone formation occurred at a high rate (>70%), mainly as ectopic bone in the SG I tests in mouse cranial connective tissues, and largely as bone augmentation in rat cranial bones in the SG II experiments when x-tHyA contained both nHAp and BMP. The prepared x-tHyA + nHAp + BMP SG material can be used as an injection-type osteoinductive bone-forming material. Sub-periosteum injection was expected.

## 1. Introduction

In dental implantology, bone formation is often desired in patients whose implants cannot be firmly placed due to shallow and narrow jaw bones [1]. Sinus lift or socket lift with autogenous bones and/or alloplasts (granules) is often performed to enlarge the areas of the bone that receive implants [2]. Treatment with an alloplast without bone collection is preferred as a remedy for patients [3]. Incisions and sutures are inevitable when placing granules of beta-tricalcium phosphate [4] and apatite [5,6]. However, alloplastic granules often spill out of implanted areas [7], causing infection problems. Therefore, the use of less invasive injection-type bone forming materials without suturing is expected [8].

In dental implantology, injection-type bone substitute materials are rarely used. Meanwhile, in orthopedic surgery, self-setting apatite-based bone cement has been used to treat vertebral compression fractures [9]. Hydrogels—such as polyethylene glycol [10], chitosan [11], alginate [12], hyaluronic acid (HyA) [13,14,15], and gelatin [16,17]—which are often coupled with calcium phosphate (i.e., apatite or tricalcium phosphate) [18] and growth factors, such as bone morphogenetic protein (BMP) [19], have been studied as injectable bone forming materials. However, most have not been used routinely in dental practice. New injectable biomaterial systems are expected to be developed in implantology.

HyA is a natural polysaccharide composed of D-glucuronic acid and N-acetylglucosamine [20] and is a component of the extracellular matrix of most connective tissues that exhibit excellent biocompatibility when applied to the human body [21]. Depending on the processing method, HyA materials can be prepared in the form of sponges, hydrogels, or injectable gels [20]. In cosmetics, HyA is often used to eliminate nasolabial folds [22]. HyA is also used for joint fluid supplementation [23], eye operation [24], wound recovery [25], and soft tissue restoration [26].

Due to its chemical structure, HyA is a hydrophilic polymer and can be characterized by a fast degradation rate (e.g., for 3–5 days) [20]. HyA-based materials have been intensively assessed for biomedical applications due to their excellent biocompatibility, biodegradability, and chemical modification [20]. HyA requires a chemical cross-link for more than a month in vivo [20,27]. HyA can be cross-linked by chemical modification and the use of an appropriate cross-linker, while HyA has been chemically modified with hydrazide [28], amino or aldehyde functional groups [29], and methacrylate groups [30] to form stable cross-link networks [20,27]. Another important approach is the thiol modification of the side chains of HyA (Figure 1a) and cross-linking by a di-functional cross-linker (Figure 1b) [31,32]. Hystem^®^—a cross-linkable thiol-modified hyaluronic acid (tHyA)—was developed in the USA for biomedical research, and is claimed to be capable of transplanting cells and/or slowly releasing growth factors [33,34,35,36,37]. This material has been followed by several rival clinical products, such as Restylane Lyft^®^ [33,38], and has not been thoroughly examined as an injection-type bone-forming material [39].

HyA is not osteoconductive, while BMP is a strong bone-forming growth factor [40]. Adding a growth factor and its carriers can render HyA-based materials osteoinductive and osteoconductive [39,40]. Nano-hydroxyapatite (nHAp) has been reported to be an osteoconductive, bio-absorbable, and carrier material, while larger hydroxyapatite blocks and granules are more inert, less bio-absorbable, and less protein adsorbed [41]. BMP can be bound to and slowly released from nHAp, sustaining long-term bone-forming activity [40,42]. We previously reported that injected x-tHyA + nHAp + BMP sol–gel (SG) successfully caused ectopic bone formation in the back subcutaneous and thigh muscles of mice by endochondral ossification [39]. As a next step, we believed that it was necessary for clinical application to check the bone-forming capability of x-tHyA + nHAp + BMP SG in the cranial osseous area of living animals.

The materials considered for bone augmentation—namely, bone grafts and substitute materials—have wide variations in the type and use method. Briefly, these materials can be classified into naturally derived materials (autografts, allografts, and xenografts), synthetic materials (hydroxyapatite, beta-tricalcium phosphate, calcium phosphate, bioactive glasses [43,44], metals, and polymers), composite materials (e.g., HyA/nHAp), growth factor-based materials, and materials with infused living osteogenic cells. Xenografts contain bovine bone, collagen, HyA, and silk [45]. Materials can also be classified based on several attributes. According to the source, they can be classified into two groups: biological (e.g., HyA) or non-biological. By chemical composition, they are metals and alloys, ceramics, or polymers (e.g., natural polymers, such as collagen and HyA, and synthetic polymers, such as polyetherether-ketone, polyethylene glycol, polylactide, and polycaprolactone). Due to their material consistency, they are three-dimensional (3D) printable, implantable solids, injectable (e.g., HyA), or adhesive. Additives may contain stem cells and bioactive agents (e.g., BMP). They may be composed of nanografts in the form of nanosized tubes, fibers, sheets, crystals, and cages [46]. The base material studied (x-tHyA) is an injectable natural-origin polymeric SG material.

First, we characterized un-cross-linked and cross-linked thiol-modified HyA (ux-tHyA and x-tHyA, respectively) by Fourier transform infrared spectroscopy (FTIR), scanning electron microscopy (SEM), hyaluronidase dissolution tests, and thermogravimetry (TG) coupled with differential scanning calorimetry (DSC). We performed two experiments to investigate the use of nHAp. We observed direct binding between nHAp and protein using confocal laser scanning microscopy and performed protein release tests in saline solution from a mixed gel of x-tHyA and nHAp.

Second, the main purpose of this investigation was to prepare an injection-type bone-forming material using x-tHyA, nHAp, and BMP and examine its usefulness in animal experiments. We examined the bone-forming ability of x-tHyA + nHAp + BMP (i) using x-tHyA + BMP with and without nHAp (SG I series) in mouse cranial subcutaneous connective tissues, and (ii) employing x-tHyA + nHAp with and without BMP (SG II series) on rat cranial bones by X-ray approaches and histological observations. The final objective of this study was the rapid development of a novel injectable bone-forming material system using existing HyA material (x-tHyA), nHAp, and BMP and to search for information on the technology and supporting materials necessary to realize this original purpose. The novelty of this study lies in the open publication of the bone-forming capability by the combined use of nHAp and BMP in x-tHyA, which could lead to its future direct-wide therapeutic use in dentistry and medicine. In particular, we have examined the usefulness of nHAp in biomedical applications [47,48]. The final intended use of the investigated materials is its direct subperiosteal injection to achieve alveolar bone augmentation for dental implant placement and insertion of denture.

## 2. Materials and Methods

### 2.1. Material

A commercial cross-linkable hyaluronic acid kit (Hystem^®^ Kit, 12.5 mL, Part Number GS1004, Sigma-Aldrich, St. Louis, MO, USA) consisting of hyaluronic acid possessing –SH functional groups (Glycosil^®^) (Catalog Number GS220, ESI BIO, Alameda, CA, USA), thiol-reactive polyethylene glycol diacrylate (PEGDA) cross-linker (Extralink Lite^®^, Catalog Number GS3008, ESI BIO), and degassed (DG) water (Calatog Number GS241, ESI BIO) (Figure 2). Other drugs and materials used were commercial recombinant human/mouse/rat (CHO cell-derived) bone morphogenetic protein-2 (BMP) (R&D Systems, Catalog Number 355-BM, Minneapolis, MN, USA), nHAp with a mean diameter of 40 nm (nano-SHAp, SofSera, Tokyo, Japan), and a microorganism-derived HyA (HYALURONSAN HA-SHY, average molecular weight = 1,500,000–3,900,000, Kewpie Co., Tokyo, Japan) (HyA control). The nHAp particles were autoclaved before mixing.

### 2.2. Material Constitution and Experimental Design

Gycosil^®^ was re-constituted with DG water to form an uncross-linked sol (ux-tHyA). The Hystem^®^ hydrogel (x-tHyA) was prepared by adding the cross-linker PEGDA (Extralink Lite^®^) with DG water to the ux-tHyA sol, following the manufacturer’s instructions. Table 1 and Table 2 show the material composition and experimental design of this study, respectively. The raw x-tHyA material used was Gycosil^®^ (sponge) (tHyA). The addition of water to Gycosil^®^ produced sol (ux-tHyA). The addition of a cross-linker to ux-tHyA created x-tHyA. x-tHyA was both sol and gel, depending on the timing of cross-linker mixing completion. Before and after 20 min of mixing, x-tHyA was a sol and a gel, respectively. Therefore, x-tHyA is called a SG material. During the sol stage, it can be injected and gelled in vivo over time (Table 1). In this study, three types of experiments were carried out: (a) material characterization of HyA; (b) characterization of the use of nHAp; and (c) animal experiments using x-tHyA (Table 2).

### 2.3. Material Characterization of Control HyA, ux-tHyA, and x-tHyA

For material studies, the HyA control was dissolved in distilled water at 0.25 wt% concentration. Samples of HyA control sol, ux-tHyA sol, and x-tHyA gel were frozen at −80 °C and freeze-dried for 12 h. To differentiate the chemical and physical properties of the three HyA materials, four in vitro experiments were performed using dried samples.

#### 2.3.1. FTIR

FTIR equipped with an attenuated total reflectance attachment (Nicolet6700, Thermo Fisher Scientific, Waltham, MA, USA) (using a single reflection diamond, a refractive index of 2.38 at 1000 cm^−1^ and angle of incidence of 45°) was used to characterize the chemical structures of the dried HyA control, ux-tHyA, x-tHyA, and cross-linker PEGDA. During the measurement, the resolution was 4 cm^−1^, the wavenumber range was 4000–400 cm^−1^, and the number of scans was 10. The OMNIC software (Thermo Fisher Scientific, Waltham, MA, USA) was used to collect and process the IR spectra. For all recorded FT-IR spectra, corrections for noise from the diamond attachment and CO_2_ were performed manually.

#### 2.3.2. SEM

SEM (SU8010, Hitachi High-Tech Corp., Tokyo, Japan) was used at 15 kV to morphologically compare dried HyA control, ux-tHyA, and x-tHyA. The dried HyA samples were glued to carbon tape, placed on an aluminum stub and plasma coated with OsO_4_ using an OPC60A (Filgen, Nagoya, Japan). The thickness of OsO_4_ was 30 nm.

#### 2.3.3. Hyaluronidase Dissolution Tests

In the hyaluronidase dissolution tests, each sample (1.0 mg) (n = 6) of dried HyA control, ux-tHyA, and x-tHyA samples were dissolved in 0.01 wt% hyaluronidase solution (Code 18240-36, Nacalai Tesque, Kyoto, Japan) diluted in distilled water (0.5 mL) in a 1.5 mL microtube, and had been placed in a constant temperature bath, and kept at 37 °C. The dissolution condition was visually inspected every 6 h, and the time to complete disappearance (min) was recorded.

#### 2.3.4. TG/DSC Thermal Analyses

TG/DSC was performed on each 1 mg sample (dried HyA control, ux-tHyA, and x-tHyA) (n = 1), using specialized equipment (STA409C, Netzsch, Selk, Germany) so that the thermal stability of x-tHyA could be scaled with reference to those of HyA control and ux-tHyA. The experimental conditions for TG/DSC were as follows: atmospheric gas, nitrogen; gas flow rate (sample), 50 mL/min; gas flow rate (reference), 20 mL/min; temperature range, 20 °C to 550 °C; heating rate, 10 °C/min; sample holder, open aluminum crucible; reference, alumina (6.8 mg); TG resolution, 5 μg; and DSC resolution, <1 μW.

### 2.4. Characterization of the Use of nHAp

#### 2.4.1. Observation of Binding between nHAp and Protein

Fluorescein isothiocyanate (FITC)-labeled bovine type I collagen (1 mg/mL) (#4001, Chondrex, Inc., Redmond, WA, USA) (0.5 mL) was mixed with nHAp (0.6 mg) containing PBS (−) x2 buffered solution (0.5 mL) in a 1.5 mL microtube and held at 4 °C for 12 h (nHAp*FITC-Collagen (+)). The same solution containing nHAp particles was prepared by mixing 0.01 M acetic solution (0.5 mL) and PBS (−) ×2 buffered solution (0.5 mL) without FITC-labeled collagen (nHAp*FITC-Collagen (−)). Both solutions with nHAp were centrifuged at 56× *g* (rotation radius = 50 mm, rotation speed = 1000 rpm) for 1 min. The supernatants were discarded, and the bottom pellets were resuspended in PBS (−) solution (300 μL). The solutions were then transferred to glass dishes (25 mm in diameter), stood still for 1 h, and the powders on the bottoms of the two glasses were observed with a confocal laser scanning microscope (A1RHD25, Nikon Co., Tokyo, Japan). The measurement conditions were an excitation wavelength of 488 nm and an emission wavelength of 500–550 nm.

#### 2.4.2. Accelerated Protein Release Tests from x-tHyA Containing nHAp

For accelerated protein release tests of x-tHyA with nHAp in an aqueous environment, x-tHyA sol (6.25 mL) was produced using Gycosil^®^ (50 mg), DG water (4 mL), bovine serum albumin standard (BSA) (2 mg/mL) (1 mL) (Thermo Scientific, Rockford, IL, USA), and PEGDA cross-linker (1.25 mL) with nHAp powder (10 mg) (x-tHyA*nHAp (+)), and each sol was poured into four 1.5 mL microtubes, followed by gelation. Sols without nHAp were prepared in the same proportion and separated into four tubes (x-tHyA*nHAp (−)). Phosphate buffered saline solution (−) at a volume of 1 mL made from PBS tablets (#T900, Takara Bio, Kusatsu, Shiga, Japan) was added to gel samples in tubes after gelation, which had been stored at 37 °C in a constant temperature bath and the solution was collected 1, 3, 5, and 7 days after gelation and stored at −20 °C until measurements while new saline solutions (1 mL) were added to gels 1, 3, and 5 days later. The quantities of BSA eluted in solution were measured using a Pierce BCA protein assay kit (Thermo Scientific, Rockford, IL, USA) with four samples with two repetition measurements (n = 4 × 2) so that the protein release kinetics of x-tHyA and the protein binding/releasing trend of nHAp in x-tHyA could be visualized in a time-dependent manner.

### 2.5. Preparation of SG I and SG II Materials

Material preparation was performed aseptically on a clean bench.

#### 2.5.1. SG I Sample

The preparation protocol for the SG I samples (x-tHyA + BMP ± nHAp) was as follows. First, BMP (50 μg) was re-constituted with a 4 mM HCl solution (0.25 mL in total) with 0.5 wt% bovine fetal albumin standard (fraction V) (Production no. DK59769, Thermo Scientific Pierce, Waltham, MA, USA) as adjuvant and diluted in DG water (5 mL in total). Second, freeze-dried Glycosil^®^ (50 mg) was re-constituted in the sol state with DG water and BMP (5 mL) on a vibrating mixer (Mild Mixer PR-12, Tokyo, Japan) for 12 h at 20 °C (Liquid A). The Extralink Lite^®^ was diluted with DG water (1.25 mL) (Liquid B). Third, Liquid B was mixed with Liquid A to obtain a viscous solution (Figure 3a). Membrane filtration (0.22 μm) was used for sterilization. Half of the sol was manually mixed with nHAp (50 μg) with a plastic spatula in a 35 mm culture dish (test samples; SG I*nHAp (+) = x-tHyA + BMP + nHAp), while the other half was unmixed (control samples; SG I*nHAp (−) = x-tHyA + BMP). Both SGs were injected with a needle and syringe (Figure 3b), followed by gelation for approximately 20 min at 37 °C. The injection volume of the SG I samples was set at 200 μL in the sol stage. The BMP content of each injected SG I*nHAp (+) was 1.6 μg.

#### 2.5.2. SG II Sample

In the case of the SG II samples (tHyA + nHAp ± BMP), a mixture of ux-tHyA and nHAp was first prepared, followed by the addition of a BMP-containing HCL solution and a cross-linker solution to form the test samples (SG II*BMP (+) = x-tHyA + nHAp + BMP)—using mixing proportions of x-tHyA, nHAp, and BMP—similar to those of SG I samples. Control samples (SGII*BMP (−) = x-tHyA + nHAp) were also produced using the HCL solution without BMP. Before animal studies, test and control SG II in sol stage (50 μL each) were poured into a Teflon ring (inner hole diameter = 4 mm, outer hole diameter = 6 mm, and thickness = 2 mm) placed on a glass slide and set at 20 °C. The amount of BMP in each SG II*BMP (+) gel sample was 0.4 μg.

### 2.6. Animal Experiments

The study protocol was approved by the Ethics Committee on Animal Research of Iwate Medical University (#30-001).

#### 2.6.1. SG I Sample

Twenty 10-week-old male BALB/cAJcl mice (CLEA Japan) were used. Groups of two to three mice were housed in separate cages and provided with a standard diet and water ad libitum. Before injection, the skull hairs of the mice were removed mainly using an electric shaver. Under anesthesia with a mixture of isoflurane (3 vol%) and oxygen (0.5 L/min) gas generated by a carburetor (IV-ANE; Olympus, Tokyo, Japan), the test and control SG I samples were injected in sol stages (0.2 mL) (SG I*nHAp (+) and SG I*nHAp (−), respectively) into the cranial subcutaneous tissue of mice (Figure 4), respectively (n = 10 each) with the use of a 24-gauge needle, and fed for a duration of 8 weeks. The animals were sacrificed by CO_2_ inhalation.

#### 2.6.2. SG II Sample

Four male Wistar rats weighing 340 ± 16 g (mean ± SD) were used. All rats were housed in separate cages (two rats per cage) with a standard diet and water ad libitum. Under anesthesia with a mixture of isoflurane and oxygen, the centers of the rat calvariae were shaved and sterilized with 10% povidone iodine, followed by a local injection of anesthetic (0.2 mL, 2% lidocaine with 1:80,000 epinephrine). Then, the periosteum flaps were elevated and the cranial bone was exposed with a scalpel and bone forceps. Two rats were used for test and control SG II gels, respectively. Three tests and control SG II gels (SG II*BMP (+) and SG II*BMP (−), respectively) in Teflon rings (Figure 5a) were placed directly on the cranial bones of a rat (Figure 5b) and tightly closed with soft nylon (Softretch 4-0; GC, Tokyo, Japan). Most of the periosteum was detached from the cranial bone during the operation. Eight weeks after surgery, all rats were sacrificed by CO_2_ inhalation.

#### 2.6.3. X-ray Analyses

We evaluated new bone formation by SG I samples in the cranial subcutaneous tissues of mice using a 3D microcomputed tomography (micro-CT) system (eXplore Locus; GE Healthcare, Wilmington, MA, USA). The ossification trends in the operation areas of the cranial bones of rats using SG II samples were evaluated using a soft X-ray apparatus (M60, Softex, Tokyo, Japan).

#### 2.6.4. Histological Observations

Skulls or cranial skins of rats used for SG I and SG II samples were collected after feeding for 8 weeks with a diamond saw (MC-201 Microcutter; Maruto, Tokyo, Japan) or scissors, fixed in 10% neutral buffered formaldehyde equivalent (Mildform, Wako Chemical, Osaka, Japan) for 4 weeks at 4 °C, and decalcified in 0.5 wt% ethylene diamine tetra-acetate solution (Decalcifying solution B, Wako Chemical, Osaka, Japan) for 4 weeks at 4 °C. The cranial regions were cut from the skulls, treated with graded alcohol and xylene, and embedded in wax. The wax specimens were then cut into five μm sections using a microtome (IVS-410, Sakura Finetek, Tokyo, Japan). The sections of the slides were stained with hematoxylin and eosin (HE), followed by histological observations using fluorescence microscopy (All-in-one BZ-9000; Keyence, Osaka, Japan).

In Figure 6, the evaluation method of ossification of cranial bone for SG II samples based on line measurements is schematically illustrated. In Figure 6a, the newly formed bone by SG II is indicated in the boxed area on the left side. The dotted box area in Figure 6a is enlarged in Figure 6b. Bone formation activities were scaled by the length of the bone zone on multiple perpendicular lines. For example, the bone length in line 4 was determined by the sum of 4a, 4b, and 4c. The bone-forming activities in the Teflon ring (4 mm wide) were assessed using lines at 100 μm intervals (40 lines). As a control, the boxed area of the sham bones was selected (Figure 6c), and the bone length in lines at 100 μm intervals (40 lines) inside the boxed area in Figure 6c was measured (Figure 6d). The bone-forming activities of the SG II samples were evaluated with respect to those of the sham control bones.

### 2.7. Statistical Analyses

Free statistical software (EZR version 1.55, Saitama Medical Center, Jichi Medical University, Saitama, Japan) [49] was used for nonparametric tests, such as Fisher’s exact, Kruskal–Wallis and Mann–Whitney U tests. The null hypothesis was rejected at *p* < 0.05.

## 3. Results

### 3.1. Material Characterization of HyA Control, ux-tHyA, and x-tHyA

#### 3.1.1. FTIR

Figure 7a shows the FTIR charts of dried (i) HyA control, (ii) ux-tHyA, (iii) x-tHyA, and (iv) PEGDA cross-linker. Figure 7b shows the chemical structures of ux-tHyA with detailed side chains [35], PEGDA cross-linker, and x-tHyA. The three materials (HyA control, ux-tHyA, and x-tHyA) had similar IR absorption peaks, such as OH or NH peaks at approximately 3300 cm^−1^, C=O peak at approximately 1600 cm^−1^, and C-O peak at 1045 cm^−1^ due to the common basic HyA structures. In contrast, two test tHyA samples (ux-tHyA and x-tHyA) possessed a weak SH stretching peak at approximately 2600 cm^−1^ while control HyA lacked this peak, which reflected thiol modification of both ux-tHyA and x-tHyA. The effect of PEGDA cross-linking on x-tHyA is ambiguous. The relative peak height intensity at approximately 2900 cm^−1^ due to C-H of x-tHyA was higher than that of ux-tHyA, resulting from adding PEGDA elements with abundant C-H bonds to ux-tHyA.

#### 3.1.2. SEM Observations

Figure 8a shows an SEM photomicrograph of a dried HyA control. It was highly porous and fibrous. Figure 8b,c show those of the dried ux-tHyA and x-tHyA, respectively. Microscopically, ux-tHyA had a loose and porous structure, while x-tHyA had a denser and flat structure.

#### 3.1.3. Hyaluronidase Dissolution Tests

Figure 9 shows the results of the hyaluronidase dissolution tests of three dried HyA samples. The mean dissolution time of x-tHyA was 2763 min, significantly longer than those of the HyA control and ux-tHyA of < 110 min (*p* < 0.05). This means that cross-linking of x-tHyA resulted in a significant increase in hyaluronidase dissolution time compared with that of uncross-linked ux-tHyA. The dissolution time of ux-tHyA (104 min) was longer than that of the HyA control (17 min) (*p* < 0.05).

#### 3.1.4. TG/DSC

Figure 10a,b show the TG and DSC curves of the TG/DSC thermal analyses of the dried HyA control, ux-tHyA, and x-tHyA, respectively. The TG curves revealed that all three HyL samples maintained their weight up to approximately 220 °C, followed by gradual weight loss, while the loss rates of ux-tHyA and x-tHyA were smaller than that of the HyA control (Figure 10a). The DSC curves clarified that both ux-tHyA and x-tHyA had broad distinctive endothermic peaks in the temperature range of 200–400 °C, while the HyA control lacked a peak (Figure 10b). The two peak endothermic temperatures of the DSC curves are indicated by the arrows.

### 3.2. Characterization of the Use of nHAp

#### 3.2.1. Observation of Direct Binding between nHAp and Protein

Figure 11a–c show the bright field, fluorescence, and overlay images of nHAp particles without FITC-labeled collagen (nHAp*FITC-Collagen (−)), respectively. It was confirmed that the nHAp itself was not fluorescent and that the nHAp particles tended to agglomerate in the saline solution. Figure 11d–f show those of nHAp particles mixed with FITC-labeled collagen(nHAp*FITC-Collagen (+)). It became evident that FITC-labeled type I collagen was strongly bound to nHAp powders, causing strong fluorescent reflections. This is a direct proof of the binding between nHAp and the protein. In the latter case, nHAp agglomeration appeared to be suppressed by the presence of collagen.

#### 3.2.2. Accelerated Protein Release Test

Figure 12 indicates the results of the accelerated protein release test of x-tHyA with and without nHAp (x-tHyA*nHAp (−) and x-tHyA*nHAp (+)) in a saline solution. The cumulative protein release amounts versus the original BSA quantity (wt%) in the x-tHyA gels were plotted as a function of the elution period (days). Figure 12 presents two important observations. The BSA protein was retained but slowly released from x-tHyA without nHAp (x-tHyA*nHAp (−)) in a time-dependent manner. However, the amount of protein eluted (wt%) from x-tHyA*nHAp (−) reached 100% after incubation for 7 days. In the case of the addition of nHAp to x-tHyA (x-tHyA*nHAp (+)), the amount of BSA protein released slightly decreased in all elution periods compared to those of x-tHyA*nHAp(−) (*p* < 0.05; Mann–Whitney U test). These results implied that nHAp was strongly bound to the BSA protein in x-tHyA and retarded the release of protein from x-tHyA into a saline solution, and x-tHyA*nHAp(+) could release the protein for more than 7 days.

### 3.3. Animal Studies

#### 3.3.1. Animal Experiments with SG I

When using the SG I test material (SG I*nHAp (+)), cranial ossification was observed in subcutaneous connective tissue in 7 out of 10 mice, while the control SG I materials (SG I*nHAp (−)) did not form bone in 10 mice, and most Teflon rings were dropped. According to Fisher’s exact test, it can be stated that test SG I caused statistically significant ossification in the cranial subcutaneous connective tissues (success rate = 70%) of mice compared to the control SG I (success rate = 0%) (*p* < 0.05) (Table 3).

The ossification of SG I (SG I*nHAp (+)) is unique. Figure 13a,b show the front and side micro-CT images of one mouse cranial bone 8 weeks after SG I*nHAp (+) was injected into the cranial connective tissue. Cranial bone formation was evident. Figure 13c shows four sliced rectangles inside the newly formed bone on the existing cranial bone shown in Figure 13b. Figure 13d schematically indicates the state of bone existence on four slices of Figure 13c. Most of the ossification occurred inside subcutaneous connective tissues such as ectopic bone, while bone augmentation and connection with preexisting cranial bone were quite limited and partial (e.g., cases (ii) and (iii) of Figure 13d). Figure 13e shows a low-magnification HE-stained image of the cranial subcutaneous tissue of a mouse 8 weeks after injection of SG I (SG I*nHAp (+)) material. Figure 13f shows a high-magnification HE image of the yellow rectangle area in Figure 13e. The residual material (RM) appears to be light purple. The periosteum (PS) originated in the deep interior of the cranial bone and thickened. Island-like small new bone (NB) fragments exist above the existing cranial bone (EB), both of which are anchored to the periosteum (PS). This continuation of bone between EB and NB bridged by PS was evidence of partial bone augmentation of preexisting bone.

In contrast, the control SG I (SG I*nHAp (−)) material did not induce new bone (NB) formation in mouse cranial bone. Figure 14a shows a front micro-CT image of one mouse cranial bone 8 weeks after control SG I was injected into the cranial connective tissue. Figure 14b shows a low-magnification HE-stained image of the cranial subcutaneous tissue of a mouse 8 weeks after injection of SG I control material (SG I*nHAp (−)). Figure 14c shows a high-magnification HE image of the yellow rectangle area in Figure 14b. Residual material (RM) was present above the cranial bone, but did not cause bone formation.

#### 3.3.2. Animal Experiments with SG II

When using test SG II (SG II*BMP (+)) in six Teflon rings on rat cranial bones, one ring fell off and five rings remained (83% survival and bone formation rate). Figure 15a,b show three test SG II samples inside Teflon rings placed on one rat cranial bone after 8 weeks of placement and the corresponding soft X-ray images, respectively. Test SG II induced additional bone formation inside the Teflon ring. Figure 15c shows a low-magnification HE-stained image of the cranial bone of a rat 8 weeks after the placement of test SG II (SG II*BMP (+)). Figure 15d,e show higher magnified HE images the sites of which are indicated by yellow rectangles (i) and (ii) in Figure 15c, respectively. Test SG II (SG II*BMP (+)) produced NB both on the preexisting cranial bone (EB) (Figure 15c,d) and around the RM that appears in light purple (Figure 15c,e). The boundary between the NB and EB was observed, as indicated by the * marks in Figure 15d. This case of bone formation can be termed bone augmentation. It was also characteristic that the periosteum (PS) thickened and appeared to prelude to NB. Around the residual test SG II*BMP (+) (RM), both the NB and PS existed as island forms in Figure 15e far away from the preexisting bone. This type of bone may be called ectopic bone in connective tissue. Figure 16a shows a low-magnification HE image of the sham rat cranial bone. Figure 16b shows a high-magnification HE image of the yellow rectangle area shown in Figure 16a. No bone increment was observed above the preexisting bone. Figure 16c shows the tool-box graph of multiple line-scaled cranial bone lengths using test SG II (SG II*BMP (+)) and sham cranial bones. In the test SG II material, the bone length in the cranial region was enlarged with a statistically significant difference compared to the sham bones (*p* < 0.05).

When using the control SG II (SG II*BMP (−)), two rings were lost; two rings were filled with dermal tissue and a slight ossification was found inside the two rings on the cranial bone. Control SG II did not induce stable cranial bone formation within Teflon rings. Figure 17a,b show the skin of a rat with control II gel (SG II*BMP (−)) in two Teflon rings after 8 weeks of placement and the corresponding soft X-ray image, respectively. The Teflon rings were separated from the cranial bone and embedded in the cranial skin. NB did not form inside the rings, while weak X-ray opacity was observed inside the rings, analogous to the surrounding skin tissues (Figure 17b). Figure 17c shows a low-magnification HE-stained image of the interior of the Teflon ring 8 weeks after placement of control gel II (SG II*BMP (−)). Figure 17d shows the magnified HE-stained image of the yellow-dotted rectangle in Figure 17c. Inside the ring, dermal tissues (DM) were filtered and filled the inner space of the Teflon ring, while residual material (control gel II) (RM) was minimally observed. Inside the dermal tissues, inflammatory cells were widely infiltrated, blood vessels existed, and no bony structures were found.

## 4. Discussion

Cross-linkable HyA (x-tHyA) was confirmed by FTIR to be thiol-modified and cross-linked using a PEGDA cross-linker (Figure 7). The physical and chemical properties of uncross-linked and cross-linked thiol-modified HyA (ux-tHyA and x-tHyA) appear to be caused by the characteristic base structure and polymerization reaction. Glycosil^®^ and Extralink Lite^®^ covalently bond to each other like Lego blocks. When mixed, Extralink’s acrylate reacts with the thiol groups of the Glycosil^®^ components by click chemistry (Michael addition reaction). Crosslinks form in trans (e.g., Glycosil molecules can link to neighboring Glycosil molecules). In addition, given Glycosil’s large molecular weight and ability to adopt semiflexible random coil configurations, it is likely to loop back and bind to the cis. The final clear, transparent, viscoelastic hydrogel formed at physiological pH and temperature in approximately 20 min and was greater than 98% water. This timeframe allows an investigator to customize the hydrogel with drugs to load and deliver the mixture through a cannula [33]. The molecular weight of ux-tHyA must be significantly greater and more hydrophobic than that of the HyA control. Therefore, ux-tHyA absorbed less water, resulting in a porous but denser structure compared to the HyA control after freeze-drying (Figure 8); and it was relatively heat resistant, comparable to x-tHyA (Figure 10). It also became evident that by cross-linking, x-tHyA became very resistant to hyaluronidase (Figure 9), had a dense and flat surface upon freeze-drying (Figure 8), and was relatively heat-durable when heated at temperatures higher than 200 °C (Figure 10). This plain surface might be attributed to the formation of stronger film-like structures of x-tHyA after freeze-drying. The freeze-dried structures of HyA samples might reflect the molecular and cross-linked structures of gels with abundant water, while an increase in molecular weight leads to a decline in water content [50]. Water evaporation from the lower molecular structures of HyA materials (e.g., HyA control) during freeze-drying might create fibrous and porous freeze-dried structures. However, the fibrous and porous structures of the HyA control and ux-tHyA were not used in the animal studies of this study. Gels (x-tHyA) prepared by mixing ux-tHyA and water were used as base materials and applied to the cranial area of murines.

It was confirmed that the protein (collagen type I labeled with FITC) was directly bound to nHAp (Figure 11), which made the study to consider adding nHAp and BMP to x-tHyA meaningful. Two experimental results using test SG I and SG II indicate that x-tHyA + nHAp + BMP SG material was osteoinductive in the murine cranial areas (Figure 13, Figure 15 and Figure 16). The dual use of nHAp and BMP is a prerequisite for successful bone formation using x-tHYA (Table 3). The lack of nHAp or BMP led to the failure of reliable bone formation in both SG I and SG II studies. For bone formation using x-tHyA, BMP is primarily necessary for direct bone induction [51], and nHAp is also indispensable as a carrier to absorb and slowly release BMP to maintain the subsequent time required bone formation [52,53,54,55]. Proteins have been reported to be absorbed and slowly released by nHAp through electrostatic interactions [56]. It was reported that x-tHyA maintained and could release BMP for up to 4 weeks [33], but our experimental results (Figure 12) contradict this report. x-tHyA itself (x-tHyA*nHAp(−)) retained the BSA protein for only 7 days (1 week). In contrast, with the addition of nHAp, x-tHyA (x-tHyA*nHAp(+)) retained BSA protein over 7 days, and its dissolution was capable of continuing for 2–3 weeks (Figure 12). This delayed protein release has been considered beneficial for bone regeneration [52,53], which generally requires a long period of time (more than a month) [57]. Before animal studies, it is important to unveil these material characteristics of x-tHyA because few studies have been published [33].

Although the protein type and quantities between BSA in the elution tests and BMP used in the animal studies differed considerably, the phenomena observed in the previous test results (Figure 12)—especially the delayed protein elution trend by nHAp—might be applicable to the latter animal studies. We used three mixed proteins due to their availability and cost performance. The approximate molecular weights of BMP, BSA, and collagen type I are 26, 66, and 300 kDa, respectively [58]. The size of the protein is a key element for apatite binding. Although apatite can bind to all three proteins, a protein with a lower molecular weight is considered to bind more easily to nHAp. Such proteins with lower molecular weights could be loaded and released from nHAp for a longer period. Increased molecular weight of the protein may present a 3D conformational hindrance [59] to bind with nHAp. The surface charge may be another important factor [60]. HAp has positive and negative charges, which are beneficial for binding to oppositely charged proteins (including almost all charged proteins). Therefore, the findings observed in BSA and type I collagen for nHAp could be applicable to BMP for nHAp with intensified levels. The use of BSA instead of BMP for slow-release studies of carrier materials has been common and is considered a standard protocol [61]. BSA has been loaded as a protein model drug in many studies [62].

The consideration of animal studies is as follows. As mentioned, x-tHyA was biocompatible and largely bio-absorbed but remained in vivo for 8 weeks and did not cause adverse effects such as inflammation, immunological reactions, or fibrous tissue encapsulation [63].

We used mice and rats for the SG I and SG II experiments, respectively, because mice were more cost-effective and easier to handle for simple injection, and Teflon ring insertion was practically limited to the size of the rat cranial zone. During feeding, the animals actively moved and touched the injected sites, leading to movement and loss of the injected material in vivo. Consequently, bone formation at the target location may be hindered. The use of the cap and band could stabilize and protect the injected materials in future studies [64].

Martinez-Sanz et al. [19] reported a successful bone augmentation in mandibular bone in rats by injection of gels consisting of self-formulated cross-link-type HyA, nHAp, and BMP. They precisely injected their biomaterials subperiosteally into the innate mandibular diastema (an inactive site) and increased bone volume proportional to the dose of BMP applied. However, subperiosteal injection of gel into the cranial bone is generally quite difficult [65], and success reports have been rare [66].

In this study, we performed the injection of test SG I into the subcutaneous tissues above the cranial bone of mice and placement of SG II directly on the exposed cranial bone of rats, lacking full cover of the periosteum, respectively, and achieved partial success in additional cranial bone formation. During the injection of test SG I, the tip of the needle might slightly break the periosteum, leading to limited contact between the SG and the cranial bone, and most of the injected substances were located within the cranial connective soft tissue. Ectopic bone formation [39] occurred predominantly in cranial connective tissues, and cranial bone augmentation of preexisting bone by the membranous ossification mechanism [67] occurred in limited amounts (Figure 13). The former type of bone formation is undesirable for future clinical use. The size and morphology of the newly formed bones by test SG I varied between samples. We report a case of ectopic bone formation by SG I in Figure 13. Ectopic bone formation has been reported to significantly alter the size and morphology, making morphological analysis quite difficult [68,69]. When placed directly on the cranial bone using test SG II inside the Teflon ring, the periosteum was detached from the cranial bone, and most folded to the periphery during the first operation. During the healing process, the periosteum might recover cranial bone, while test SG II was in the process of biodegradation. We adopted a new morphological evaluation method for bone formation using SG II materials containing BMP based on line measurements (Figure 6). The bone formation level of the test SG II materials was not compared with that of the control SG II materials because the latter did not produce reliable numerical data. We used the heights of the sham bones as a control for comparison with test SG II and confirmed that the test SG II materials induced considerable new bone formation (Figure 16). Two mixed modes of bone formation were observed. One major part was bone augmentation from preexisting bone, while the other minor part was ectopic bone in the connective tissue around the remaining material above the preexisting bone (Figure 15). The latter bone was also undesirable.

It should be stated that the BMP application site determines bone quality, such as in ectopic and orthotopic models [70]. When applied to an osseous site, the osseous bone may be formed. However, when applied to soft tissue, such as connective tissues and muscle, BMP might cause ectopic bone formation. Prudent site selection is necessary for tissue engineering when using injectable scaffolds with BMP for bone augmentation. The appropriate dose of BMP is also an important factor for successful bone augmentation. The amount of BMP used in this study was comparable to that used in other studies [17,19,40,52,68,70]. Precise subperiosteal injection of SG in sol stage containing moderate dose of BMP is anticipated by developing a new surgical technique to achieve 100% bone augmentation from preexisting bone [71].

An alternative approach could be to fill box-type bone defects with SG material coupled with a covering of membranous material [72], although it deviates from injection-only treatment. Bone formation using only injected sols is a fascinating technique for future clinical dentistry due to its ease of handling and less invasive treatment, as mentioned previously [8,73]. In situ gelation of the poured sol is desirable for long-term position stability [74]. It is hoped that the set x-tHyA gel that is currently still soft and viscous will be hardened to increase the position stability.

## 5. Conclusions

Within the limitations of this study, using an SG material—thiol-modified hyaluronic acid (x-tHyA)—the following conclusions were obtained:(1)Instrumental analyses (FTIR, TG/DSC, and SEM) confirmed the presence of thiol modifications and characteristic cross-linking reactions. The cross-linked structure appeared to result in a considerably long hyaluronidase dissolution time and a denser/plain microstructure upon freeze-drying.(2)We confirmed that nHAp is directly bound to proteins and could play an important role as a growth factor carrier to hold and release proteins for a long period, which is beneficial for sustaining the time required bone formation.(3)BMP is a strong bone-forming growth factor. However, its application must be localized to osseous sites for clinically significant bone augmentation. BMP easily causes undesirable ectopic bone formation when applied to connective soft tissues. It is highly anticipated that a technique will be developed for the precise subperiosteal injection of bone-forming SG materials.(4)The combined use of nHAp and BMP for x-tHyA succeeded in bone formation at the rate of 70−83%, achieving a relatively high positive record. The dual use of nHAp and BMP is vital for successful bone formation.(5)Because the materials used (x-HyA, nHAp, and BMP) are widely available, it is highly desirable to develop a new injectable bone-forming system consisting of these materials for widespread clinical use soon.

## Figures and Tables

**Figure 1 polymers-14-05368-f001:**
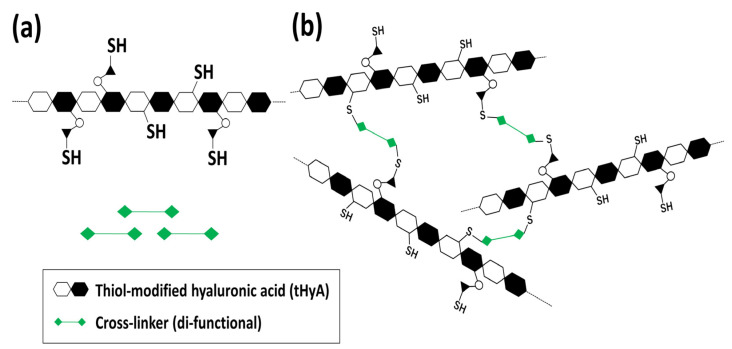
(**a**) Thiol modification of side chains of HyA, (**b**) cross-linking of tHyA by a di-functional cross-linker.

**Figure 2 polymers-14-05368-f002:**
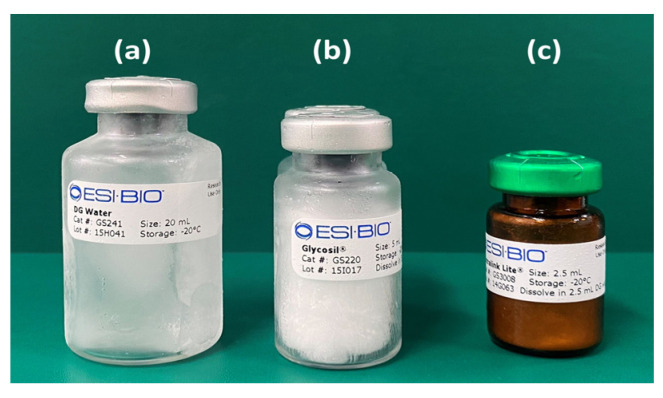
A cross-linkable hyaluronic acid kit, Hystem^®^, consisting of: (**a**) degassed (DG) water; (**b**) hyaluronic acids possessing –SH functional groups (Glycosil^®^); and (**c**) thiol-reactive polyethylene glycol diacrylate (PEGDA) cross-linker (Extralink Lite^®^).

**Figure 3 polymers-14-05368-f003:**
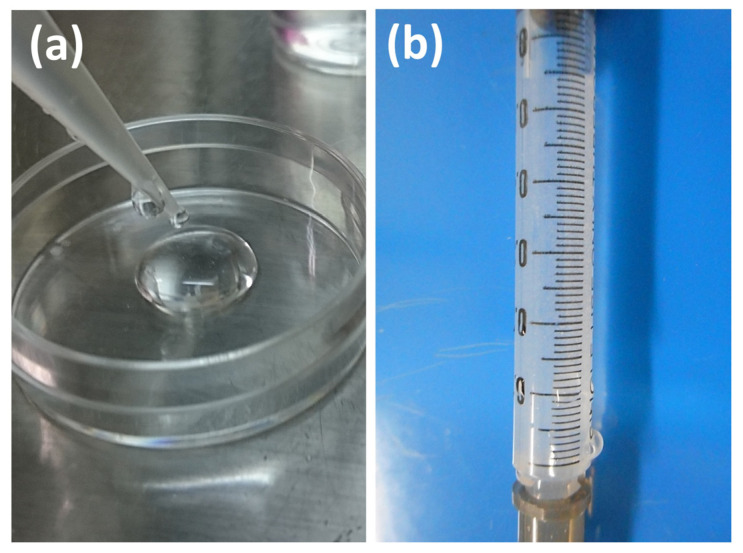
(**a**) SG I*nHAp (−) = x-tHyA + BMP in sol stage; (**b**) SG I*nHAp (+) = x-tHyA + BMP + nHAp in sol stage in a syringe.

**Figure 4 polymers-14-05368-f004:**
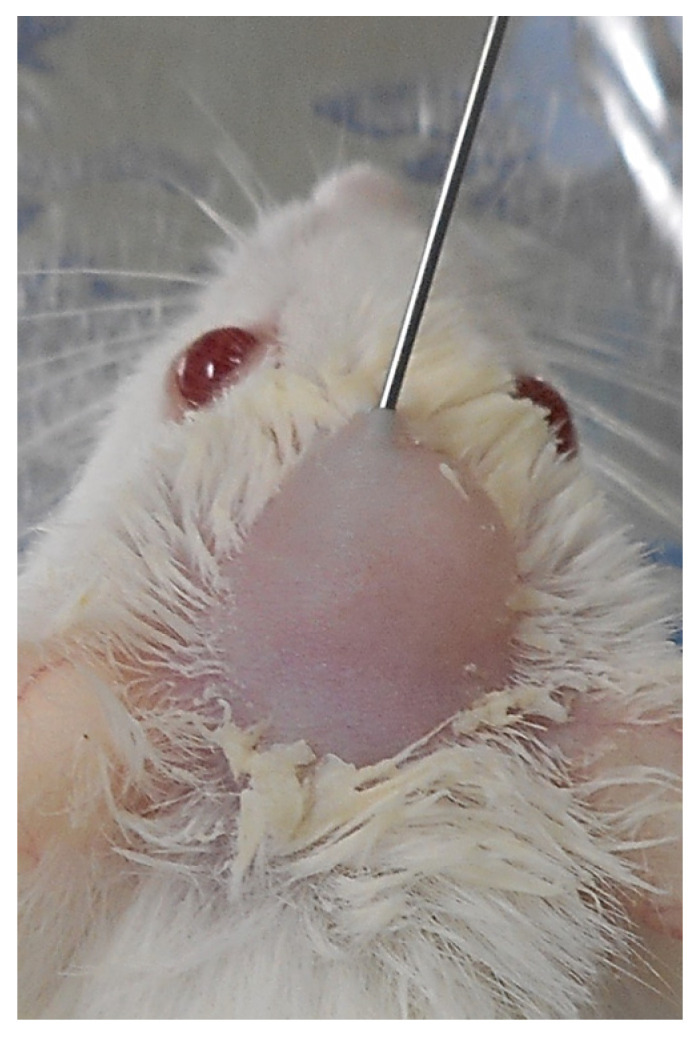
Subcutaneous injection of SG I*nHAp (+) = x-tHyA + BMP + nHAp sample in sol stage into the mouse cranial area.

**Figure 5 polymers-14-05368-f005:**
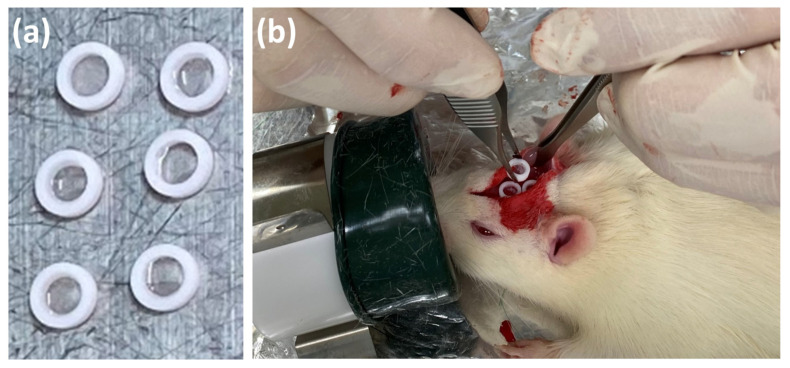
(**a**) Prepared SG II*BMP (+) = x-tHyA + nHAp + BMP samples gelled in Teflon rings; and (**b**) implantation of SG II*BMP (+) samples on exposed rat cranial bone.

**Figure 6 polymers-14-05368-f006:**
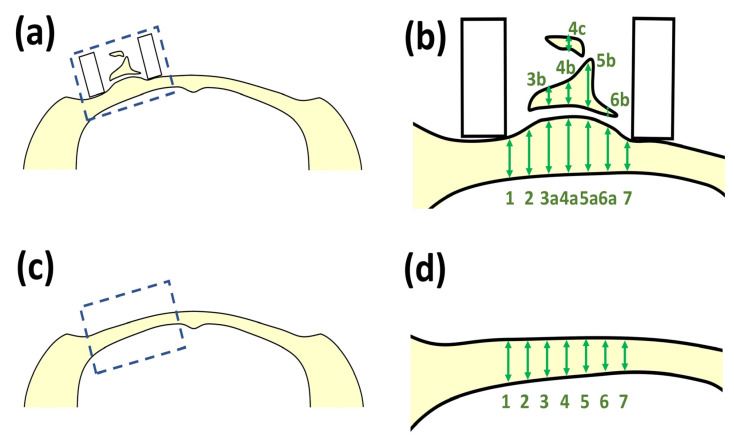
(**a**) Newly formed bone by SG II sample on the left side cranial bone; (**b**) enlarged bone area of the dotted box area in Figure 6a. Note: bone formation activities were scaled by the length of the bone zone on multiple perpendicular lines; (**c**) the sham rat cranial bone; and (**d**) the enlarged bone area of the dotted box area in Figure 6c. Note: Bone thickness was scaled by the length of the bone zone on multiple perpendicular lines.

**Figure 7 polymers-14-05368-f007:**
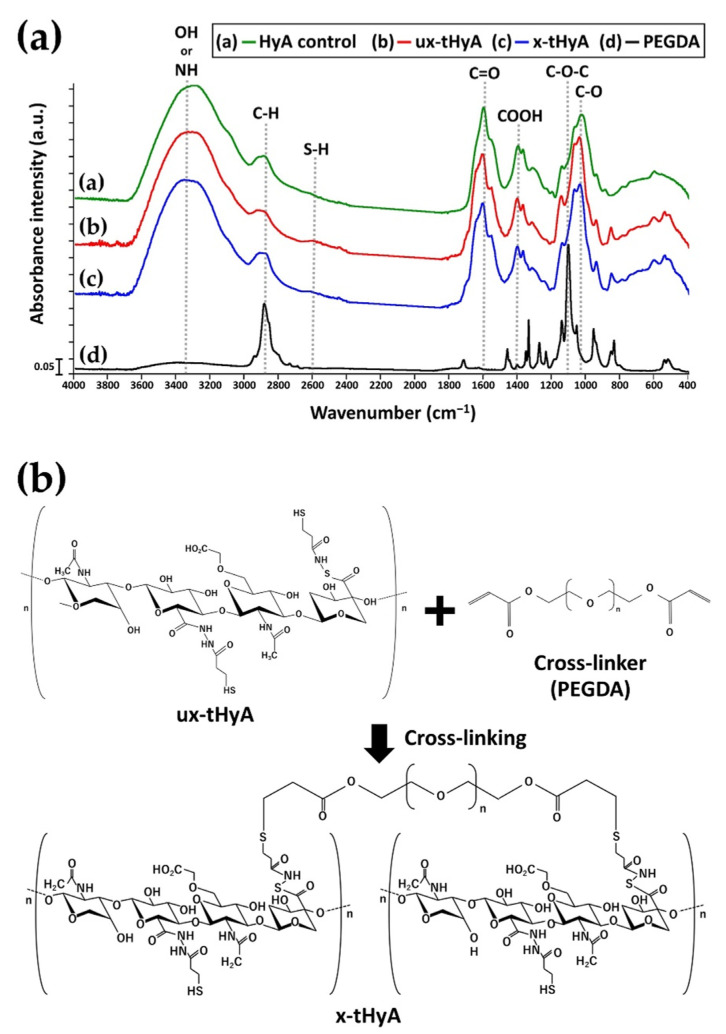
(**a**) FTIR charts of dried (i) HyA control, (ii) ux-tHyA, (iii) x-tHyA, and (iv) PEGDA cross-linker. (**b**) Chemical structures of ux-tHyA with detailed side chains, PEGDA cross-linker and x-tHyA.

**Figure 8 polymers-14-05368-f008:**
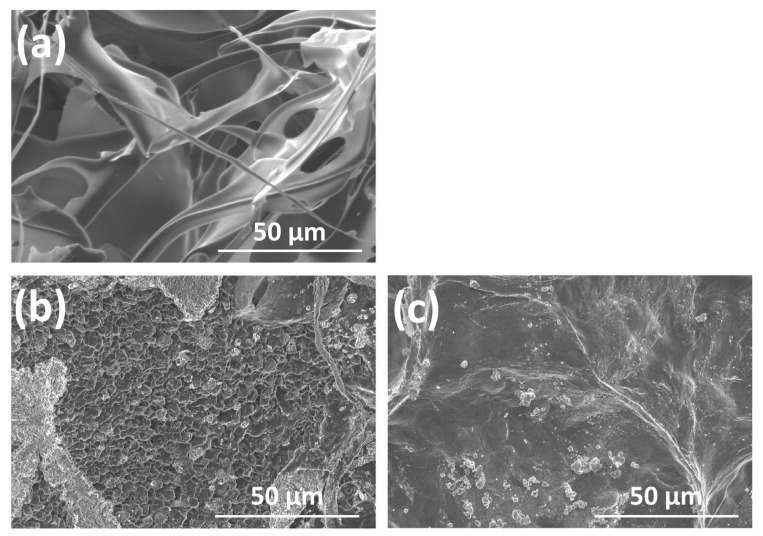
SEM photo-micrographs of dried (**a**) HyA control, (**b**) ux-tHyA, and (**c**) x-tHyA.

**Figure 9 polymers-14-05368-f009:**
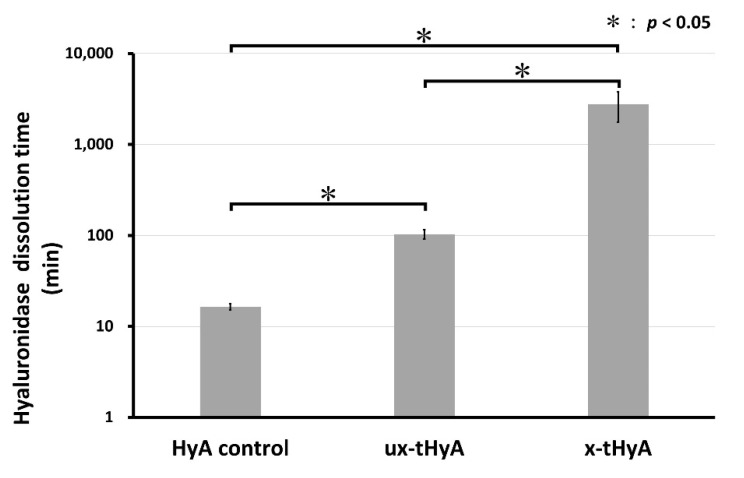
Results of the hyaluronidase dissolution test of three dried HyA samples—HyA control, ux-tHyA, and x-tHyA. Standard deviation error bars were added to graphs. The statistical analysis between two graphs combined by horizontal bar was carried out by Kruskal–Wallis test. *: *p* < 0.05.

**Figure 10 polymers-14-05368-f010:**
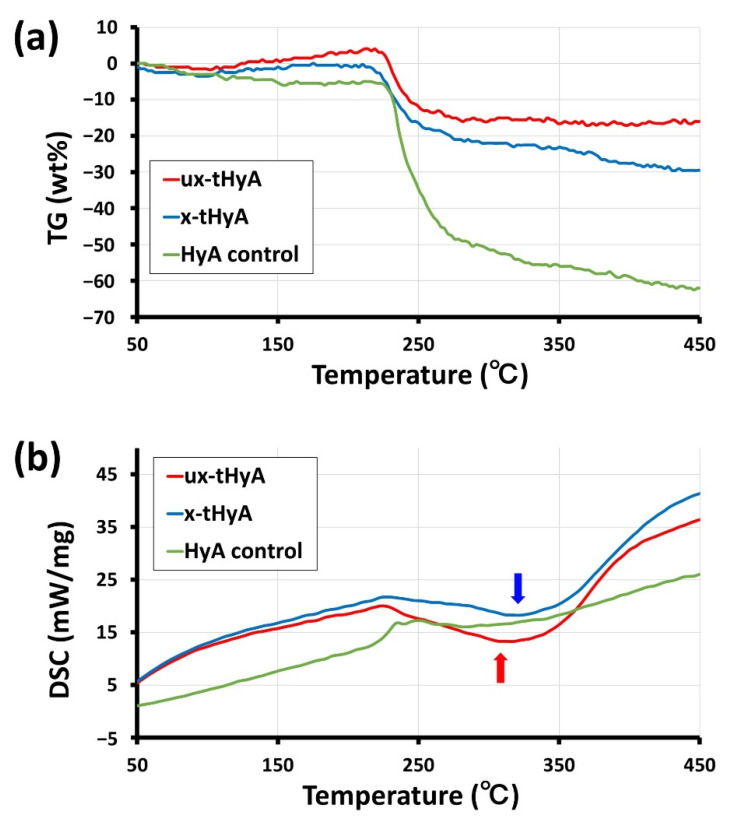
(**a**) TG and (**b**) DSC charts of TG/DSC thermal analyses of dried HyA control, ux-tHyA, and x-tHyA. Note: Arrows indicated peak temperatures of DSC endothermic curves of two samples.

**Figure 11 polymers-14-05368-f011:**
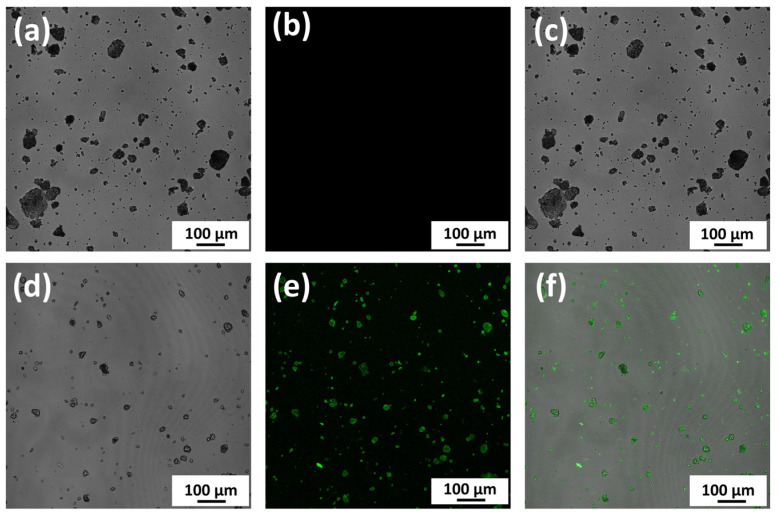
(**a**) The bright field, (**b**) fluorescence, and (**c**) overlay images of nHAp particles without FITC-labeled collagen (nHAp*FITC-Collagen (−)); (**d**) bright field, (**e**) fluorescence, and (**f**) overlay images of nHAp particles mixed with FITC-labeled collagen (nHAp*FITC-Collagen (+)).

**Figure 12 polymers-14-05368-f012:**
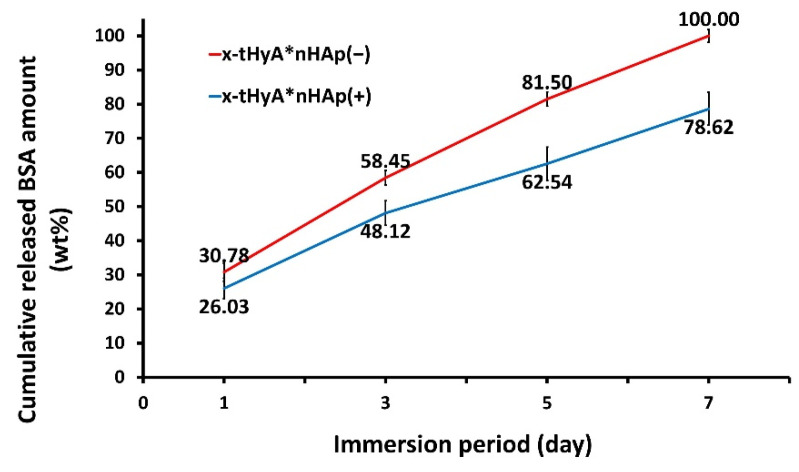
Results of the accelerated protein (BSA) release test of x-tHyA with and without nHAp (x-tHyA*nHAp (−) and x-tHyA*nHAp (+)) in saline solution.

**Figure 13 polymers-14-05368-f013:**
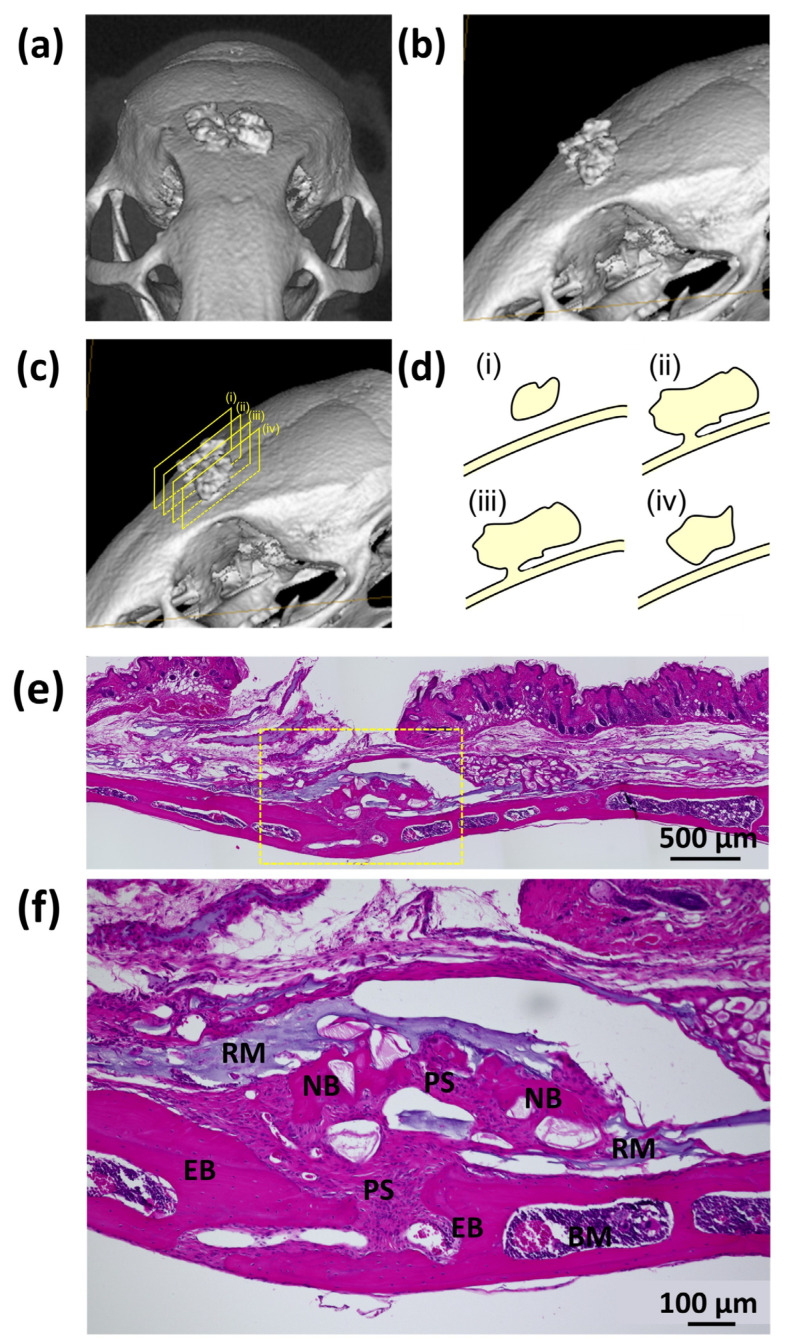
(**a**) Front and (**b**) side micro-CT images of one mouse cranial bone 8 weeks after test SG I (SG I*nHAp (+)) was injected into the cranial connective tissue; (**c**) four sliced rectangles inside the newly formed bone on the cranial existing bone of Figure 13b; (**d**) schematical indication of bone existence states on four slices of Figure 13c; (**e**) the low magnified HE-stained image of the cranial subcutaneous tissue of a mouse 8 weeks after injection of SG I (SG I*nHAp (+)) material; (**f**) the high-magnified HE image of the yellow rectangle area of Figure 13e. Note: RM, residual material; PS, periosteum; NB, new bone; EB, existing bone; BM, bone marrow.

**Figure 14 polymers-14-05368-f014:**
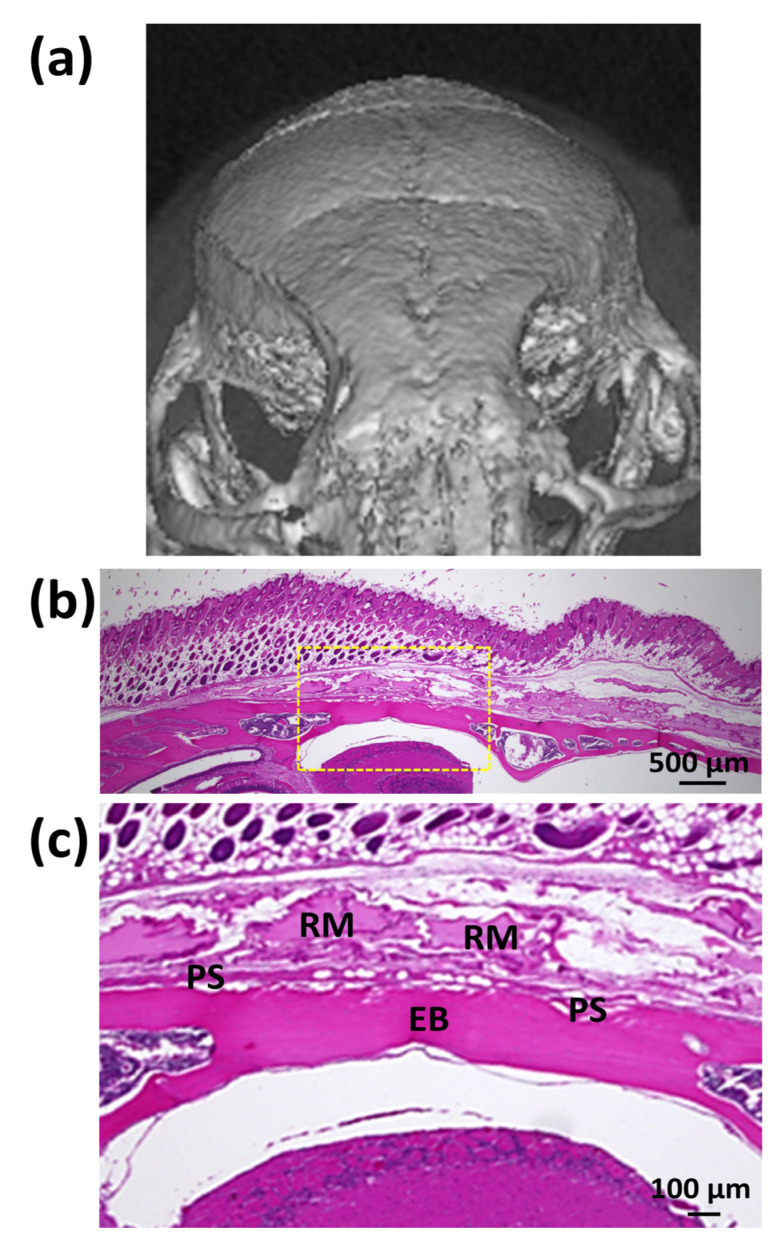
(**a**) Front micro-CT image of one mouse cranial bone 8 weeks after control SG I (SG I*nHAp (−)) was injected into the cranial connective tissue; (**b**) the low magnified HE-stained image of the cranial subcutaneous tissue of a mouse 8 weeks after injection of SG I (SG I*nHAp (−)) material; and (**c**) the high-magnified HE image of the yellow rectangle area of Figure 14b. Note: RM, residual material; PS, periosteum; EB, existing bone.

**Figure 15 polymers-14-05368-f015:**
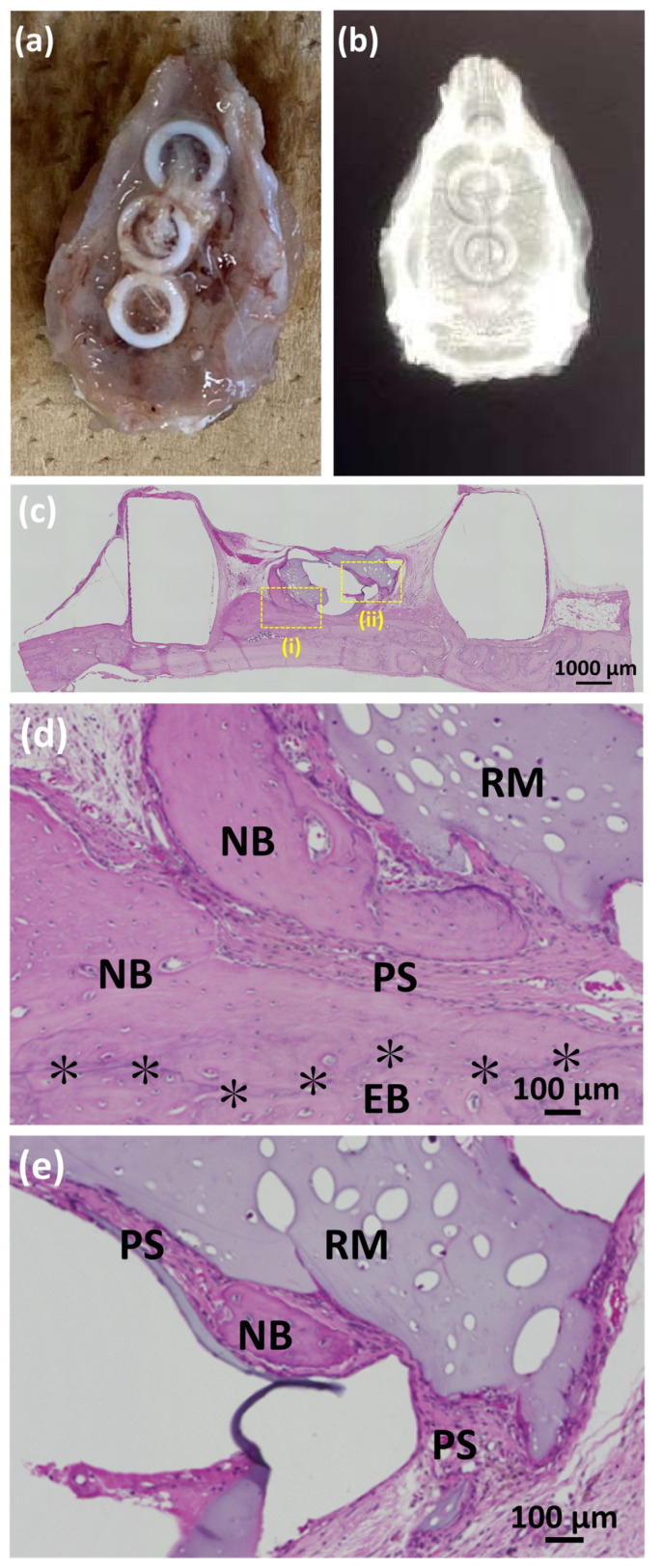
(**a**) Photo and (**b**) soft X-ray image of three test SG II samples (SG II*BMP (+)) inside Teflon rings placed on one rat cranial bone after 8 weeks of placement; (**c**) the low magnified HE-stained image of the cranial bone of a rat 8 weeks after the placement of test SG II (SG II*BMP (+)); (**d**) higher magnified HE images of the yellow rectangle (i) of Figure 15c; and (**d**,**e**) higher magnified HE images of the yellow rectangle (ii) of Figure 15c. Note: RM, residual material; PS, periosteum; NB, new bone; EB, existing bone. Note: * indicates the border between NB and EB.

**Figure 16 polymers-14-05368-f016:**
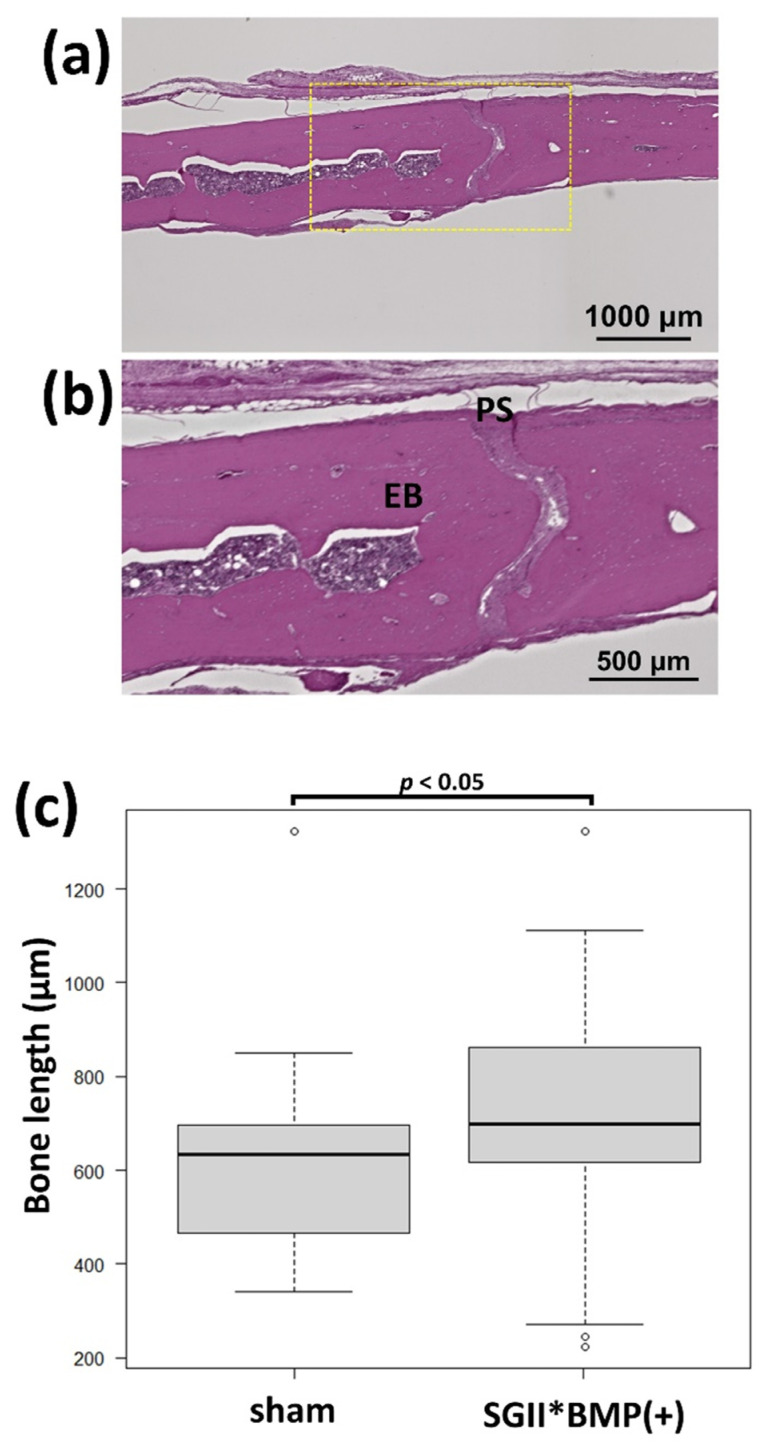
(**a**) The low-magnified HE image of the sham rat cranial bone, (**b**) the high-magnified HE image of the yellow rectangle area in Figure 16a, and (**c**) the tool-box graph of multiple line-scaled bone length of cranial area bone using test SG II (SG II*BMP (+) = x-tHyA + nHAp + BMP) and sham cranial bones. The statistical analysis was performed by Mann–Whitney U test.

**Figure 17 polymers-14-05368-f017:**
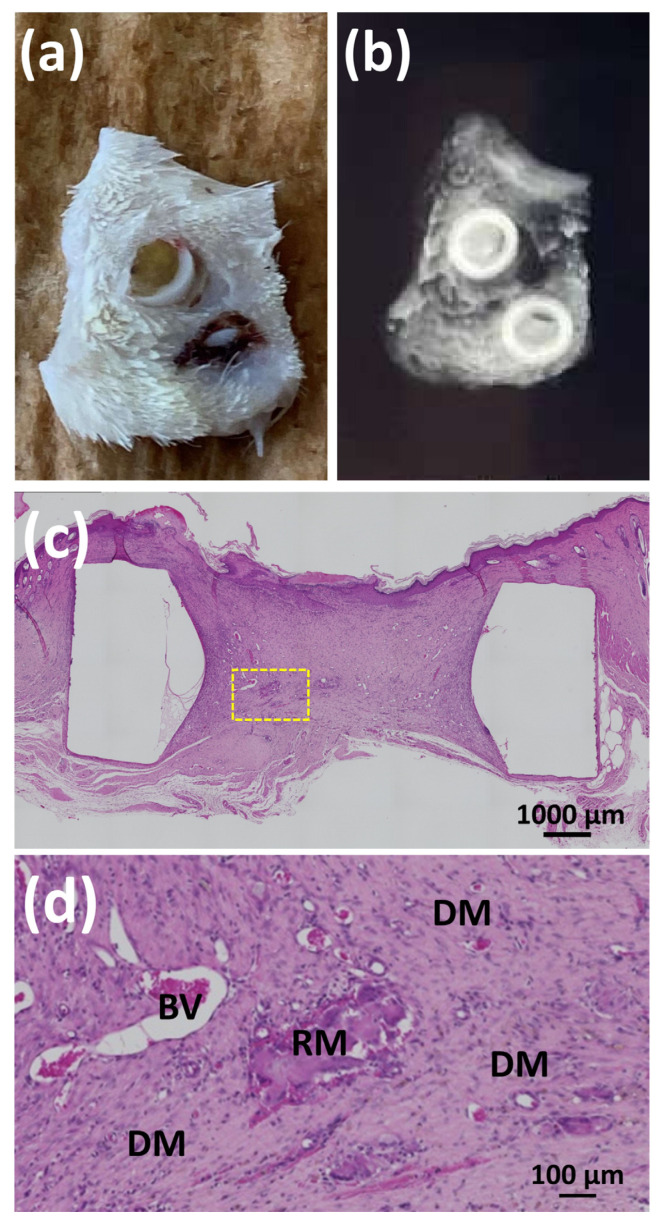
(**a**,**b**) Photograph of the skin of a rat with control II gels (SG II*BMP (−)) in two Teflon rings after 8 weeks of placement and the corresponding soft X-ray image, respectively; (**c**) low-magnification HE-stained image of the interior of the Teflon ring 8 weeks after the placement of control gel II (SG II*BMP (−)); and (**d**) magnified HE-stained image of the yellow-dotted rectangle in Figure 17c. Note: RM, residual material; DM, dermal tissues; BV, blood vessels.

**Table 1 polymers-14-05368-t001:** Materials explanation.

Materials	Material Constituent	State
HyA control	HYALURONSAN HA-SHY	Powder
tHyA	Glycosil^®^	Sponge body
ux-tHyA	Glycosil^®^ + DG water	Sol
At this time, BMP, and nHAp were added to ux-tHyA		
x-tHyA	Glycosil^®^ + DG water + Extralink Lite^®^	Sol (up to 20 min) Gel (after 20 min)
Dried HyA control	Freeze-dried (HyA control + water)	Sponge body
Dried ux-tHyA	Freeze-dried ux-tHyA	Sponge body
Dried x-tHyA	Freeze-dried x-tHyA	Sponge body

**Table 2 polymers-14-05368-t002:** Design of materials and experiments.

(a)				
**Dried Samples**	**Material Characterization of HyA**			
	**FTIR**	**SEM**	**Hyaluronidase Dissolution Tests**	**TG/DSC**
HyA control	●	●	●	●
ux-tHyA	●	●	●	●
x-tHyA	●	●	●	●
(b)				
**Samples**	**Characterization of the Use of nHAp**			
	**Observation of** **Protein Binding**		**Accelerated Protein** **Release Tests**	
nHAp*FITC-Collagen (±)	●			
x-tHyA SG*nHAp (±)			●	
(c)				
**Samples**	**Animal Experiments**			
	**Direct Injection** **into Cranial Area**		**Set in a Teflon Ring and Placed on Cranial Bone**	
SG I*nHAp (±)	●			
SG II*BMP (±)			●	

Note: sol–gel (SG). ● means the experiment conducted. * means “and”.

**Table 3 polymers-14-05368-t003:** Bone-forming numbers in the cranial subcutaneous connective tissues of mice 8 weeks after injection of 10 control SG I*nHAp (−) and 10 test SG I*nHAp (+) samples, respectively.

Sample	Number of Ossification Confirmed	Number of Ossification Not Confirmed
Control SG I*nHAp (−) ^#^ (x-tHyA + BMP)	0	10
Test SG I*nHAp (+) ^#^ (x-tHyA + BMP + nHAp)	7	3

* means “and”. ^#^
*p* < 0.05 by Fisher’s exact test.

## Data Availability

All data are included in the manuscript.

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
