# Peer review of "Bone Formation on Murine Cranial Bone by Injectable Cross-Linked Hyaluronic Acid Containing Nano-Hydroxyapatite and Bone Morphogenetic Protein"

_polymers, 2022, doi:10.3390/polym14245368_

Round 1
Reviewer 1 Report
This study includes nano-hydroxyapatite (nHAp) and bone morphogenetic protein (BMP) to cross-linkable a thiol-modified hyaluronic acid (xHyA), to be evaluated in vivo as a convenient osteo-inductive injectable material.
1-Please include the following references related to bioactive glasses in the introduction to complete other ways to promote bone growth:
https://doi.org/10.1016/j.jnoncrysol.2020.120206
DOI: 10.1243/09544119JEIM836
2-Figure 8, must be modified, it is not possible to read it, since it is so small.
Author Response
Reply 1
Yes, we added these 2 references in the manuscript.
- Beltrán, A.M.; Alcudia, A.; Begines, B.; Rodríguez-ortiz, J.A.; Torres, Y. Porous titanium substrates coated with a bilayer of bioactive glasses. Journal of Non-Crystalline Solids 2020, 544, 120206. doi: 10.1016/j.jnoncrysol.2020.120206
- Jones, J.R.; Lin, S.; Yue, S.; Lee, P.D.; Hanna, J.V.; Smith, M.E.; Newport, R.J. Bioactive glass scaffolds for bone regeneration and their hierarchical characterisation. Proc Inst Mech Eng H 2010, 224(12), 1373-1387, doi:10.1243/09544119JEIM836
Comment 1
-Please include the following references related to bioactive glasses in the introduction to complete other ways to promote bone growth:
https://doi.org/10.1016/j.jnoncrysol.2020.120206
DOI: 10.1243/09544119JEIM836
Reply 2
Yes, Figure 7 (b), previously Figure 8 was enlarged.
Comment 2
Figure 8, must be modified, it is not possible to read it, since it is so small.
Author Response
I send a word file to you.

Reviewer 3 Report
Hachinohe with collaborators presented the study concerning development of bone-forming materials based on nano-hydroxyapatite (nHAp) and bone morphogenetic protein (BMP) to cross-linkable thiol-modified hyaluronic acid (xHyA) for biomedical applications. In this study short physicochemical characterisation and in vivo biological studies are shown. The presented data seem to be valuable and interesting for the scientific community, however, I have some major issues that authors should address before the publication process:
1. During the reading of the paper it is hard to understand exactly what materials are control, tested ones and why in some experiments authors only studied/presented the results of selected materials. The abbreviations and descriptions of the studied materials also should be consistent. Moreover, in Materials section the table with the list and description of the studied materials could be helpful.
2. The manuscript especially in the Results and Discussion section should be revised and reorganised to ease the reader to understand the idea of the presented studies with more consistency and reading flow.
3. Why did the authors use nanoHAp instead of micrometric-sized HAp powder? Please, put more emphasis on this aspect.
4. Why authors used uncross-linked microorganism derived HyA (HYALURONSAN 96 HA-SHY) as the control material? Please explain.
5. It is not clear which materials are sols and which are gels. This should be somehow clarified in the manuscript.
6. The authors stated that “Upon mixing, PEGDA’s acrylates reacted with thiol groups of uxHyA via Michael addition with thiolyne click chemistry to form xHyA [33].” That should be explained more deeply.
7. Authors presented SEM images of only two prepared materials. Why the rest is not shown?
8. As the porosity of the materials is one of the important aspect in bone formation, did the authors perform porosity/pore size studies using i.e. mercury intrusion or microCT? That would be valuable information in the manuscript as the authors stated that “uxHyA had the loose and porous structure whilst xHyA possessed denser and flat structure because of crosslinking” and “Freeze-dried xHyA had denser microstructures…”.
9. There are some commas instead of dots i.e. in line 275.
10. Authors stated that “These results implied that nHAp strongly bound to BSA protein..”. That should be evidenced and explained more.
11. Some of the figures could be merged especially in the case of in vivo studies.
12. In the discussion the authors wrote “Although the protein quantities between BSA in the elution tests and BMP employed in animal studies considerably differed…” Not only quantities but also the protein type may have an impact? Please explain.
13. The conclusions should be revised with more quantitative data. Please revise.
14. There are some grammatical and spelling mistakes throughout the text – please revise.
Considering the above I recommend the manuscript to be published in Polymers after major revision.
Author Response
I send you a Word File.

Round 2
Reviewer 3 Report
The authors responded to my concerns, and they revised the Manuscript in detail. The changes made by the authors significantly improved the quality of the paper. Thus, I recommend the paper to be accepted in the present form.